# Identification of Peanut *AhMYB44* Transcription Factors and Their Multiple Roles in Drought Stress Responses

**DOI:** 10.3390/plants11243522

**Published:** 2022-12-14

**Authors:** Yonghui Liu, Yue Shen, Man Liang, Xuyao Zhang, Jianwen Xu, Yi Shen, Zhide Chen

**Affiliations:** Jiangsu Academy of Agricultural Sciences, Nanjing 210014, China

**Keywords:** peanut, R2R3-MYB TF, *AhMYB44* overexpression, drought stress, transgenic plant

## Abstract

MYB transcription factors (TFs) comprise a large gene family that plays an important role in plant growth, development, stress responses, and defense regulation. However, their functions in peanut remain to be further elucidated. Here, we identified six *AhMYB44* genes (*AhMYB44-01/11*, *AhMYB44-05/15*, and *AhMYB44-06/16*) in cultivated peanut. They are typical R2R3-MYB TFs and have many similarities but different expression patterns in response to drought stress, suggesting different functions under drought stress. Homologous genes with higher expression in each pair were selected for further study. All of them were nuclear proteins and had no self-transactivation activity. In addition, we compared the performances of different lines at germination, seedling, and adult stages under drought stress. After drought treatment, the overexpression of *AhMYB44-11* transgenic plants resulted in the longest root length at the seedling stage. Levels of proline, soluble sugar and chlorophyll, and expression levels of stress-related genes, including *P5CS1*, *RD29A*, *CBF1*, and *COR15A*, were higher than those of the wild type (WT) at the adult stage. While the overexpression of *AhMYB44-16* significantly increased the drought sensitivity of plants at all stages, with differential ABA content, the expression levels of the ABA-related genes *PP2CA* and *ABI1* were significantly upregulated and those of *ABA1* and *ABA2* were significantly downregulated compared with the WT. *AhMYB44-05* showed similar downregulated expression as *AhMYB44-16* under drought stress, but its overexpression in *Arabidopsis* did not significantly affect the drought resistance of transgenic plants. Based on the results, we propose that *AhMYB44-11* plays a role as a positive factor in drought tolerance by increasing the transcription abundance of stress-related genes and the accumulation of osmolytes, while *AhMYB44-16* negatively regulates drought tolerance through its involvement in ABA-dependent stress response pathways.

## 1. Introduction

Peanut (*Arachis hypogaea* L.) is widely grown around the world as a major source of plant protein and oil [1,2]. Transcription factors (TFs) play an important role in regulating the expression of target genes, and peanut TFs can influence production and quality [3]. With the completion of peanut genome sequencing [4,5], more and more peanut TFs have been identified and analyzed. For example, the overexpression of *AhWRKY75*, a member of the WRKY IIc subfamily, increases the efficiency of the ROS scavenging system and salt tolerance in peanut [6]. *AhbHLH112*, a bHLH TF in peanut, acts as a positive factor in drought stress tolerance [7]. In addition, the genome-wide systematic characterization and the expression profiles of NAC and bZIP in peanut have shown their roles in response to different stresses [8,9]. Recently, several studies have identified MYB TFs in peanut and carried out bioinformatics analyses, but their functions remain elusive [10,11].

MYB TFs can be divided into four subfamilies—1R-MYB, R2R3-MYB, R1R2R3-MYB, and 4R-MYB—according to the number and position of adjacent MYB motif repeats. The MYB motif repeat contains 50–53 amino acid residues and encodes three α-helices [12,13]. R2R3-MYB, with two MYB motif repeats, is specific to plants, and over 100 members are found in the genomes of dicots and monocots, making it the most abundant of the MYB families [14]. R2R3-MYBs play important roles in many biological processes, such as plant growth and development, secondary metabolism, stress and disease resistance, and signal transduction [15,16].

MYB44 is a typical R2R3-MYB TF and has been studied in many plants. In *Arabidopsis*, there are 125 R2R3-MYB genes, which can be divided into 22 subgroups [13]. *AtMYB44* belongs to subgroup 22 (S22), along with *AtMYB70*, *AtMYB73*, and *AtMYB77* [17]. The functions of *AtMYB44* are diverse and include regulating seed germination through the phosphorylation of mitogen-activated protein kinase [18], affecting resistance to the diamondback moth and green peach aphid by activating EIN2-affected defenses [19], and modulating disease resistance to *Pseudomonas syringae* through the salicylic acid signaling pathway [20]. In eggplant, *SmMYB44* regulates the expression of spermidine synthase, and its overexpression enhances plant resistance to bacterial wilt [21], while in potato, overexpression of *StMYB44* results in dwarfing, smaller leaves, curling, and smaller tubers, but the specific regulatory mechanism remains unclear [22]. *IbMYB44* in sweet potato affects anthocyanin synthesis by competitively inhibiting the formation of the IbMYB340-IbbHLH2-IbNAC56 complex [23,24]. In strawberry, there are three MYB44 homologous genes. Of these, the function of *FaMYB44.1* is related to anthocyanin accumulation, and *FaMYB44.2* plays a negative regulatory role in malic acid content and soluble sugar accumulation [25]. However, the role of MYB44 in peanut is still unknown.

Drought is an important environmental stress in plants, and R2R3-MYB TFs have been proved to be related to drought tolerance. For example, overexpression of *AtMYB12* and *AtMYB75* increases the accumulation of flavonoids through the regulation of transcripts of flavonoid biosynthesis genes and enhances drought tolerance in *Arabidopsis* [26,27]. As positive regulators of wax synthesis genes, *AtMYB94* and *AtMYB96* can activate cuticular wax biosynthesis in response to drought stress [28,29]. *AtMYB60* is related to stomatal movement regulation. Studies have found that, compared to controls, the stomatal opening of *myb60* mutant was repressed, resulting in reduced water loss and enhanced drought tolerance [30,31]. The expression of *AtMYB44* in plant guard cells is high and its role in response to drought tolerance is still controversial. *AtMYB44*-overexpression plants have improved drought tolerance due to the enhancement of abscisic acid (ABA) sensitivity in both germination and stomatal closure [32]. Some studies suggest that AtMYB44 negatively regulates ABA signaling by interacting with AtMYB77 to form a heterodimer under drought stress. The *myb44 myb77* double mutant shows strong drought resistance [33,34].

In this study, six MYB44 genes were identified in peanut and named according to their chromosome positions. Expression analysis showed different expression patterns in response to drought stress, suggesting their various functions. Their roles in response to drought stress were analyzed in transgenic *Arabidopsis* lines, and the characteristics of their subcellular localization, transcriptional activity, and cis-acting elements were also analyzed. The results reveal the functional characteristics of MYB44 in peanut and provide a reference for molecular breeding.

## 2. Results

### 2.1. Six AhMYB44 Genes were Identified in Cultivated Peanut

Six candidate genes were finally obtained and isolated from cv. Fenghua1 after sequence homology alignment and domain analysis and named *AhMYB44-01*, *AhMYB44-11*, *AhMYB44-05*, *AhMYB44-15*, *AhMYB44-06*, and *AhMYB44-16*, according to the chromosome location. All genes were found to have no intron regions after comparing their genomic DNA and cDNA sequences. The sequences of *AhMYB44s* were highly similar in pairs corresponding to the A-genomes (1–10) and the B-genomes (11–20). For example, *AhMYB44-01* and *AhMYB44-11* mapped to the homologs Arahy.01 and Arahy.11, respectively, and the open reading frames (ORFs) were both composed of 948 bp nucleotides encoding 316 amino acids. Sequence alignment results indicated that their sequences shared 98.3 % nucleotide identity and their translation products had only three different residues. Their isoelectric points (pI) were both 9.57, with molecular weights (MW) of 34.46 or 34.37 kDa. Two of the six *MYB44s*, *AhMYB44-05* and *AhMYB44-15*, were found to correspond to protein sequences in PeanutBase (*Arahy.HL1WF3* and *Arahy.CPTS0I*) (Table 1).

Furthermore, we cloned the 1.5 kb region of the promoter sequence of *AhMYB44s* from the genomic DNA and sequenced it (Appendix A). Analysis of putative cis-acting elements showed that many light-responsive elements were found in the promoter region, as well as the common G-box (TACGTG), MRE (AACCTAA), ACE (CTAACGTATT/GACACGTATG), Box4 (ATTAAT), I-box (GATAAGGTG), TCT (TCTTAC), GATA (AAGATAAGATT), and GT1 (GGTTAA) motifs. We particularly focused on the stress-related elements: seven stress-related elements were found in the *AhMYB44-11* promoter, eight in *AhMYB44-05*, and seven in *AhMYB44-15*, while only four stress-related elements were identified in *AhMYB44-01*, three in *AhMYB44-06*, and four in *AhMYB44-16*. All *AtMYB44s* had ARE and CGTCA motifs, which are involved in anaerobic induction and MeJA responsiveness, respectively. Notably, the MYB binding site (MBS) involved in drought induction was only found in the *AhMYB44-01/11* promoter, while ABA-responsive elements (ABREs) were present in both *AhMYB44-05/15* and *AhMYB44-06/16* but not in *AhMYB44-01/11* (Table 2), suggesting that they may play different and special regulatory roles in plant stress response.

The multiple sequence alignment indicated that all AhMYB44s contained the conserved motif of *Arabidopsis* MYB S22 (QxMxxxEVRxYM), a bHLH interaction motif, and a transcriptional repressor domain LxLxL (Figure 1A) [25]. Phylogenetic analysis indicated that AhMYB44s have a close relationship with AtMYB44 and MYB44 proteins from other species. Looking at the nine reported R2R3-MYBs in peanut [10], AhMYB44-01 was previously described as AhMYB4 and AhMYB44-16 as AhMYB3, and they showed a distant relationship with other proteins, except for AhMYB1 and AhMYB5 (Figure 1B). Moreover, the high bootstrap values of the proteins mapped to the homolog genomes indicated that there were few differences between the two homolog proteins, so they may have similar functions.

### 2.2. Expression Profiles of AhMYB44s

We analyzed the expression profiles of *AhMYB44s* to explore their possible biological functions. A heatmap based on peanut transcriptome data showed that homologous *AhMYB44* genes with high sequence similarity had almost the same transcript abundances. Moreover, *AhMYB44-05/15* had the highest relative transcript abundance in most tissues or organs, while *AhMYB44-01/11* showed low expression. Interestingly, their expression levels were downregulated very significantly in embryos during late pod development (Figure 2A: S16 and S17), as with *AhMYB44-06/16*. However, *AhMYB44-01/11* had relatively high transcript abundance in these two samples. In addition, all *AhMYB44s* exhibited similar expression patterns under different stress treatments, except drought stress. For example, they all showed upregulation of expression in response to paclobutrazol and low temperature treatments (Figure 2A: S24 and S27), while their expression levels were downregulated in leaves when treated with brassinolide or ethephon (Figure 2A: S23 and S25). Under drought stress, only *AhMYB44-01/11* presented upregulation, while the expression levels of *AhMYB44-06/16* were downregulated significantly (Figure 2A: S29).

In order to verify the above results, we analyzed the expression of *AhMYB44s* in different tissues and in response to drought stress using the quantitative real-time polymerase chain reaction (RT-qPCR) method. With high sequence similarity, it was difficult to discern the abundance of every gene transcription, and the relative expression levels were based on aggregating the abundances of homolog genes transcripts. As shown in Figure 2B, *AhMYB44-05/15* and *AhMYB44-06/16* expression decreased during embryo development, whereas the expression of *AhMYB44-01/11* increased. After drought treatment (Figure 2C), the expression levels of *AhMYB44-06/16* and *AhMYB44-05/15*—especially *AhMYB44-06/16—*were strongly downregulated, and only the expression of *AhMYB44-01/11* showed upregulation, although the upregulation multiple was not high (about 1.4 times). In summary, the results of the RT-qPCR were consistent with the transcriptomic data, indicating that these genes may play different roles in peanut embryonic development and response to drought stress.

### 2.3. Subcellular Localization and Transcriptional Activity Analyses of Three AhMYB44 Genes

*AhMYB44-11*, *AhMYB44-05*, and *AhMYB44-16*, homologous genes with higher expression in each pair, were selected for transcriptional activation and subcellular localization analysis. As shown in Figure 3, all yeast transformants could grow well in plates containing SD/-Trp medium. However, in SD/-Trp-Leu + X-α-gal medium, most of the yeast-transformed colonies failed to grow; the exception was the positive control, which grew well and turned blue. These results indicate that these AhMYB44s have no self-transactivation activity. To determine whether AhMYB44s were located in the nucleus, the AhMYB44-GFP fusion vectors driven by the CAMV35S promoter were constructed. Confocal microscopic analysis showed that all AhMYB44s were localized in the nucleus, whereas the GFP control protein was localized in the nucleus and cytosol, as expected (Figure 4).

### 2.4. Different Tolerances to Drought Stress of AhMYB44s in Arabidopsis

To determine the roles of *AhMYB44s* in plant drought tolerance, *AhMYB44-05*-, *AhMYB44-11*-, and *AhMYB44-16*-overexpression *Arabidopsis* transgenic plants were constructed, and two transgenic lines of each gene—named *5*-*, 11*-, and *16-OE1* and *5*-*, 11*-, and *16-OE2*—were selected for further study (Appendix A). The *Arabidopsis* wild type (WT) was used as a control, and we subjected the plants to drought stress at three different growth stages. In the seed germination stage, the WT and transgenic plants exhibited similar phenotypes under normal conditions but, with increasing mannitol concentration, the plants were significantly inhibited and had significantly shorter roots, especially in the 16-OEs (Figure 5). In the seedling stage, the taproots of the 11-OEs were longest in the experimental samples subjected to 300 mM mannitol treatment, while the 16-OEs still had significantly shorter roots than the WT (Figure 6B). Fresh weight data further confirmed that overexpression of *AhMYB44-16* increased the sensitivity of *Arabidopsis* plants to drought stress (Figure 6C). The ABA content of 16-OEs was significantly higher than that in the WT under normal condition. After drought stress treatment, the ABA content of all the tested samples increased, but the 16-OEs had the lowest relative ABA content (Figure 6D).

In the adult stage (1 month old), there were no clear phenotypic differences between transgenic plants and the WT. After drought treatment for 15 days, all plants showed growth inhibition and chlorosis and the 16-OEs showed the least vigor. Most *AhMYB44s* transgenic plants failed to survive after re-watering for 7 days, whereas 11-OEs showed higher survival rates than the WT (Figure 7A). In addition, we measured the levels of drought-resistance indicators (proline, soluble sugar, and chlorophyll) in WT and transgenic plants. There were no significant differences in these indicators among all experimental samples under normal conditions. After drought treatment, 11-OEs demonstrated higher proline, soluble sugar, and chlorophyll levels than the WT, while 16-OEs showed the opposite trend (Figure 7B–D). In summary, overexpression of *AhMYB44-16* significantly increased drought sensitivity, and overexpression of *AhMYB44-11* may lead to improve drought tolerance in transgenic plants. The 05-OEs, however, displayed no significant differences compared to the WT under drought stress.

### 2.5. Expression Pattern Analysis of Stress-Related Genes in Transgenic Arabidopsis Plants

To study the molecular regulatory mechanism of *AhMYB44-05*, *AhMYB44-11*, and *AhMYB44-16* in response to drought stress in transgenic *Arabidopsis*, we detected the expression levels of several ABA-related genes (*AtABI1*, *AtPP2CA*, *AtABA1*, and *AtABA2*) and stress-response genes (*AtP5CS1*, *AtRD29A*, *AtCBF1*, and *AtCOR15A*) using RT-qPCR.

As shown in Figure 8A, under normal conditions, there was no obvious difference in the expression levels of *AtABI1* and *AtPP2CA* between WT and transgenic lines, but their transcripts were significantly increased in the 05-OEs and 16-OEs after drought treatment. The expression levels of *AtABA1* and *AtABA2* in the 05-OEs and 16-OEs were higher under normal conditions and significantly decreased compared with the WT after drought treatment. In contrast, no significant differences were found between the 11-OEs and WT under any of the conditions. These results suggest that *AhMYB44-05* and *AhMYB44-16* may be closely related to the ABA signaling pathway.

As shown in Figure 8B, under normal conditions, most stress-response genes showed no significant differences among all the samples, except *AtRD29A* and *AtCOR15A* in the 11-OEs. The expression levels of all stress-response genes were clearly upregulated by drought stress, especially in the 11-OEs. Moreover, after drought treatment, *AtP5CS1*, *AtRD29A*, and *AtCOR15A* expressions were significantly lower at the transcriptional levels in the 16-OEs than those of the WT. The results showed that, although *AhMYB44-11* overexpression did not obviously affect the expression levels of ABA-related genes, it significantly increased the transcription levels of stress response-related genes under drought stress.

## 3. Discussion

The cultivated peanut is an allotetraploid crop (2n = 4x = 40, AABB), and the A and B genomes are very similar, frequently showing more than 98% DNA identity between corresponding genes [4]. In our study, six MYB44 genes were identified in the cultivated peanut, which could be divided into three pairs according to the high sequence similarity. Among the *AhMYB44* genes, *AhMYB44-05/15* possessed the longest sequence length and the highest expression level in most tissues and organs, which may be the reason why only these two genes exist in PeanutBase. Furthermore, we found that all AhMYB44s were located in the nucleus, as with most MYB proteins, and had no self-transactivation activity in yeast (Figure 3 and Figure 4), suggesting that their transcriptional activity in plants might depend on interacting with proteins, target promoter sequences, or a modification state [35]. In addition, these genes had some similar conserved motifs (Figure 1), such as a transcriptional repressor domain LxLxL (EAR motif), which was associated with active repression of several target genes [36]. For example, in *Arabidopsis*, AtMYB44 could interact with a TOPLESS-RELATED corepressor via the EAR motif to suppress PP2C gene transcription [37]. We also found that *AhMYB44s* exhibited similar expression patterns in most tissues and for most stress inductions (Figure 2). In this study, we focused on their differences.

In view of their different response patterns under drought stress, transgenic *Arabidopsis* plants overexpressing *AhMYB44s* were treated with drought stress at different stages (Figure 5, Figure 6 and Figure 7). As the only upregulated *AhMYB44* gene, the drought resistances of the 11-OEs were not very significant at the germination and seedling stages, which may have been related to their low upregulation multiple. However, at the adult stage, the 11-OEs showed stronger drought tolerance than the WT with a more resistant phenotype, and the contents of soluble sugar, chlorophyll, and proline were significantly higher. Proline and soluble sugar are common osmolytes in plants that accumulate massively in plants under abiotic stresses. A higher content means better abilities to stabilize the subcellular structure, scavenge free radicals, and buffer oxidative stress [38,39].

Furthermore, the stress marker genes, such as *AtP5CS1*, *AtRD29A*, *AtCBF1*, and *AtCOR15A*, were also upregulated (Figure 8B). Pyrroline-5-carboxylic acid synthase (P5CS) is a key enzyme in the proline biosynthetic pathway [40]. Studies have confirmed that overexpression of *P5CS1* or *P5CS2* can increase the content of proline and improve drought and salt tolerance in plants [41,42,43]. Based on these results, we speculated that the enhanced drought resistance of the 11-OEs was mainly due to the regulation of the expression of downstream genes and the accumulation of more compatible osmolytes. Moreover, the promoter of *AhMYB44-11* contained two MYB binding sites (MBSs) involved in drought induction [44], suggesting that *AhMYB44-11* may combine with other MYB TFs to regulate drought resistance in plants. More research is thus needed to reveal the in-depth regulatory mechanisms.

In contrast to *AhMYB44-11*, *AhMYB44-16*-overexpression transgenic *Arabidopsis* plants were the most sensitive to drought stress at each stage, representing a more inhibited phenotype with shorter root lengths and faster wilting, and the ABA content of the 16-OEs was significantly lower than the WT under drought stress. At the same time, *AtPP2CA* and *AtABI1*, the marker genes of negative regulation of the ABA signaling pathway, had significantly higher transcript abundance, while *AtABA1* and *AtABA2* [45,46], the important enzyme genes for the ABA biosynthetic pathway, had obvious downregulation in the 16-OEs under drought conditions (Figure 8A). All the results demonstrated that *AhMYB44-16* may be a negative regulator in drought stress through reduction of the content of ABA.

We also found that the lateral roots of 16-OEs were underdeveloped compared with the WT (Appendix A). Some MYB genes have been reported to be involved in drought response by regulating lateral root growth, such as AtMYB77, a key protein mediating crosstalk between ABA and auxin signaling in lateral root development in response to drought [47,48], and AtMYB60, which is thought to increase water-absorption capacity in the early stage of drought stress by inducing root growth [31]. In our results, the ABA content and the expressions of *AtABA1* and *AtABA2* were significantly higher in 16-OEs than the WT under normal conditions. Furthermore, the promoter region of *AhMYB44-16* contained an ABA-responsiveness element, ABRE. Therefore, we speculated that overexpression of *AhMYB44-16* could result in the increase in ABA content in transgenic seedlings, thereby inhibiting lateral root development and affecting its response to drought stress.

The expression pattern of *AhMYB44-05* was similar to that of *AhMYB44-16*. However, *AhMYB44-05* had higher transcript abundance in most peanut tissues and organs and more different types of response elements in its promoter region, indicating that *AhMYB44-05* is a wide-ranging regulator involved in peanut growth, development, and stress response. In our study, heterologous expression of *AhMYB44-05* in *Arabidopsis* did not significantly affect the phenotype and drought resistance of transgenic plants. It is possible that using the gene deletion mutant would better clarify its function.

In addition to different responses to drought stress, the expression patterns of *AhMYB44s* during seed development were also different. *AhMYB44-01/11* expression increased gradually during embryonic development, while *AhMYB44-05/15* and *AhMYB44-06/16* in the embryo became activated at the transition stage and then depressed to very low levels. The transition stage in legume seed development is the period when the embryo switches from a meristem-like tissue into a differentiated storage organ and soluble sugar content is at its peak, which is then followed by a gradual downward trend [49]. The transition stage in peanut pod development is also an important dividing point: during this stage, the peanut shell expands rapidly to the maximum value and then stops expanding gradually, while the seed begins to grow rapidly from this stage and gradually differentiates into leaf primordia [50,51]. Therefore, we speculated that *AhMYB44-01/11* play regulatory roles in the late stage of seed development, and the expression characteristics of *AhMYB44-05/15* and *AhMYB44-06/16* seem to be consistent with the variation in the soluble sugar content in peanut seed. Their regulatory roles in the seed may be related to seed sugar accumulation. In fact, *FaMYB44.2* has been proved to be a negative regulator of soluble sugar accumulation in strawberry fruit [25]. Although the sugar accumulation processes in strawberry fruit and peanut seeds are quite different [52], further research on the mechanism by which MYB44 regulates sugar content in peanut will help to reveal the differences between sugar metabolic networks in different crops. In order to verify the above assumptions, much more work is needed to reveal the complete function and regulation mechanism of MYB44s in peanut.

## 4. Materials and Methods

### 4.1. Experimental Materials

The peanut cultivar Fenghua1 (developed by the Yongshan Wan team at the Peanut Research Institute of Shandong Agricultural University) was used in our study. The seeds were grown in the fields of the experimental station of Jiangsu Academy of Agricultural Sciences. Roots, leaves, and stems from four-leaf-stage peanut plants and flowers, pegs, and embryos from different growing stages of mature peanut plants were sampled for gene expression analysis. For the drought-induced assay, we sterilized the peanut seeds and placed them in germination boxes. After 4 or 5 days, the germinated seeds were transferred into plastic pots containing Hoagland’s solution and grown in temperature-controlled incubators under a 16/8 h photoperiod at 25–28 °C. Then, the four-leaf-stage seedlings were treated with 20 % (*v/v*) PEG6000, and leaf samples were taken at 0, 2, 4, 6, 12, 24, 36, and 48 h. All the samples were frozen with liquid nitrogen and stored at −80 °C.

### 4.2. Sequence and Phylogenetic Analysis of AhMYB44s

To identify MYB44 in the peanut genome, the AtMYB44 (AT5G67300) protein sequence was used as a query for BLAST analysis against PeanutBase (http://www.peanutbase.org/, accessed on 28 June 2019) and the NCBI database (http://www.ncbi.nlm.nih.gov/, accessed on 28 June 2019). The CDD in NCBI (http://www.ncbi.nlm.nih.gov/structure/cdd/wrpsb.cgi, accessed on 28 June 2019) and SMART (http://smart.embl-heidelberg.de/, accessed on 28 June 2019) were used for inspection. Conserved domains of the sequences were obtained, and incomplete and redundant amino acid sequences were removed. Six candidate genes were obtained, and we designed primers to amplify them from cv. Fenghua1. Following the manufacturer’s protocol, the total RNA of peanut was isolated using a Plant RNeasy Mini Kit (TIANGEN, Beijing, China), and reverse transcription was performed with an Advantage RT-for-PCR Kit (TaKaRa, Dalian, China). The genomic DNA of peanut was used as a template, and the 1.5 kb promoter sequences of six candidate genes were amplified with PCR using specific primers. We used pEASY^®^-Blunt Simple Cloning Vectors (TransGen, Beijing, China) for the PCR products for further sequencing verification. The potential cis-acting elements in promoters were identified with the Plant-CARE database (http://bioinformatics.psb.ugent.be/webtools/plantcare/html/, accessed on 4 June 2020) [53].

Multiple alignments of AhMYB44s and members of the MYB S22 in *Arabidopsis thaliana* were carried out using Clustal X2.0.12 [54]. GeneDoc was used to edit and mark the alignment results [55]. The phylogenetic tree was constructed with the neighbor-joining method in MEGA5 using MYB44s from diverse plant species, nine R2R3-MYB sequences from peanut identified by Chen [10], and other MYB sequences. The full-length protein sequences are provided in Appendix A. Bootstrapping was set with 1000 replicates to assess the statistical reliability of nodes [56].

### 4.3. Expression Analysis of AhMYB44s

An RNA-Seq-based peanut transcriptome (http://peanutgr.fafu.edu.cn/Expression.php, accessed on 17 August 2021) was used to estimate the abundance of *AhMYB44s* transcripts [5] based on 29 peanut cDNA samples, including eight tissues and organs (roots, shoots, leaves, flowers, pegs, testas, pericarps, and embryos) at different development stages, and leaves subjected to various stress treatments. Utilizing the gene IDs or sequences, their fragments per kilobase of transcript per million mapped reads (FPKM) were obtained from the peanut transcriptome database, and the heat map was produced in TBtools with FPKM values [57].

Further validation of the RNA-Seq results was carried out with RT-qPCR. Total RNA and reverse transcription experiments were carried out on different samples using previously described methods. We used SYBR Green PCR Master Mix^TM^ (Perfect Real Time; Takara, Dalian, China) in an ABI QuantStudio^TM^ Sequence Detection System (Applied Biosystems, UK) for RT-qPCR. Each 20 μL reaction comprised 2 μL cDNA template, 10 μL 2×SYBR Premix, and 0.4 μL of each primer (200 nM). The reactions were subjected to the following conditions: 95 °C for 20 s followed by 40 cycles of 95 °C for 10 s, 55 °C for 20 s, and 72 °C for 40 s. The *AhACTIN11* and *AtACTIN2* genes were used as internal controls for peanut and *Arabidopsis*, respectively [58,59]. The relative mRNA ratios were calculated using the 2^−△△Ct^ method [60].

### 4.4. Subcellular Localization and Transcriptional Activation

In accordance with the results of the sequence alignment and expression analysis, three MYB44 genes—*AhMYB44-05* (XM_025842871), *AhMYB44-11* (XM_025771501), and *AhMYB44-16* (XM_025804513)—were selected for functional analysis.

For subcellular localization assays, three *AhMYB44s* coding sequences (excluding stop codons) were amplified with gene-specific primers and inserted into the multiple cloning site of the modified pSuper1300 vector, which contained overexpressing green fluorescent protein (35S::GFP) at the C-terminal. The empty pSuper1300 vector was used as a control. Next, the control and recombinant plasmids were electroporated into *Agrobacterium tumefaciens* strain GV3101 [61]. The NLS-RFP strain was used as a localization marker [62]. Then, tobacco leaves grown for about 4 weeks were transfected by infiltration using an *Agrobacterium tumefaciens* method [63] and incubated for 2–3 days (25 °C and 16/8 h). Fluorescence was observed with a confocal laser scanning microscope (Nikon, Tokyo, Japan).

The coding sequences of three *AhMYB44s* were cloned into a pGBKT7 vector (Clontech, Redwood City, CA, USA), and the recombinant vectors were transformed into yeast strain AH109 with a lithium acetate method (PT1172–1, Clontech, Redwood City, CA, USA). The yeast cells transformed with pGBKT7-53 and pGADT7-T plasmids were used as positive controls, and those only containing the pGBKT7 vector were used as negative controls. In accordance with the instructions (Clontech, Redwood City, CA, USA), all the yeast cells were grown on master plate synthetic defined medium without tryptophan (SD/-Trp) at 30 °C for 2–4 days, and the transformed colonies were plated onto selection medium without tryptophan and leucine (SD/-Trp-Leu + X-a-Gal) to test for transcriptional activity.

### 4.5. Transformation and Characterization of Transgenic Plants

The *AhMYB44-05*, *AhMYB44-11*, and *AhMYB44-16* coding sequences were amplified and built into the multiple cloning site of the pSuper1300 vector driven by the Super promoter [64]. The recombination vectors were transferred into *Arabidopsis* wild type (*Col-0*) using the *Agrobacterium tumefaciens*-mediated floral dip method [65]. The transformed lines were selected by growing them on half-strength Murashige and Skoog (MS) media containing 50 μg mL^−1^ hygromycin under similar growth conditions. Two T_3_ homozygous lines screened for each gene were used for further study, and the transcript levels of the transgenic lines were assessed using genomic PCR and semi-quantitative reverse transcription PCR.

For drought-stress assays at the seed germination stage, the seeds of wild-type and transgenic lines were surface-sterilized with 75 % (*v/v*) alcohol for 5 min and 2 % (*v/v*) sodium hypochlorite for 10 min. Then, sterilized seeds were placed on half-strength MS media and supplemented with 0, 300, or 400 mM mannitol. After germination in the dark for 3 days at 4 °C, the culture dishes were left for 8 days with a photoperiod of 23:21 °C and 16:8 h light: dark. Germination rates and root lengths were measured. For drought-stress assays at the seedling stage, after germination on half-strength MS medium for 4 days (the root length was about 1 cm), the seedlings were transferred to a new medium with 0 or 300 mM mannitol, and the root length and fresh weight of the seedlings were measured 7 days later. For fresh weight analysis, each repeat contained the weight of three seedlings.

The seedlings treated for 4 h, 12 h, 24 h, and 7 d were collected and mixed as the stress treatment samples and then combined with samples obtained under normal conditions. Their ABA content was determined with a liquid chromatography tandem mass spectrometry (LC-MS) system (Agilent technologies 1260 Infinity II). Referring to previously published methods and parameter information [66], the sample was ground into powder using liquid nitrogen and accurately weighed. Each 100 mg sample of powder was extracted with 1 mL of 80% (*v/v*) methanol, and this was repeated three times. The supernatant was combined and concentrated using nitrogen evaporation. The LC column was an Agilent Poroshell 120 EC-C18 (2.7 μm, 3.0 × 100 mm; USA).

To explore the drought tolerance of mature plants, the seedlings of the WT and transgenic lines in medium were transplanted into moistened soil. One-month-old plants were used to perform the drought treatments without watering. After 15 days of stress treatment, the leaves were sampled for proline, soluble sugar, and chlorophyll content analysis using the relevant kits from Keming Biotechnology Co. (Suzhou, China). The plants were then rewatered for 7 days and imaged. The leaves were also used for RNA extraction and stress-responsive gene expression analysis with RT-qPCR. Primers involved in the experiments are listed in Appendix A.

## 5. Conclusions

In this study, we characterized *AhMYB44s*, the homologous genes of *MYB44* in peanut. They had different expression patterns during seed development and response to drought stress. Further analysis showed that *AhMYB44-11* enhanced drought tolerance in *Arabidopsis* by increasing the transcripts of stress-response genes, while *AhMYB44-16* increased drought sensitivity through its involvement in ABA-dependent stress-response pathways. However, their different roles in seed development and the mechanism of *AhMYB44-16* affecting lateral root growth remain to be further studied. Taken together, our findings reveal the regulation characteristics of *MYB44* genes in peanut under drought stress and lay the foundation for the comprehensive analysis of the function of MYB44.

## Figures and Tables

**Figure 1 plants-11-03522-f001:**
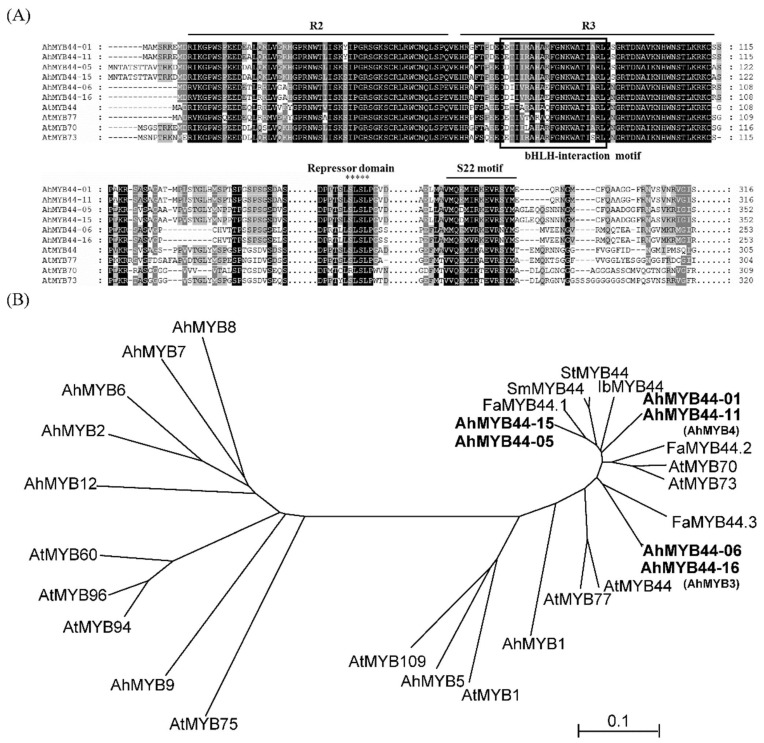
Phylogenetic analysis and multiple sequence alignment of MYB44. (**A**). Multiple sequence alignment was performed using the protein sequences of AhMYB44s and *Arabidopsis* MYB S22 (AtMYB44, AtMYB70, AtMYB73, AtMYB77). The R2 and R3 domains are labeled, and bHLH-interaction motifs were included in the R3 domain; the negative repressor motif LxLxL and the conserved domain of *Arabidopsis* MYB S22 are all labeled. (**B**). Phylogenetic analysis of MYB44s from diverse plant species and some R2R3-MYBs from *Arabidopsis* and peanut. The amino acid sequences are indicated as follows: Ah, *Arachis hypogaea*; At, *Arabidopsis thaliana*; Fa, *Fragaria* × *ananassa*; Sm, *Solanum melongena*; St, *Solanum tuberosum*; Ib, *Ipomoea batatas*. The scale bars indicate that 10 of every 100 amino acids showed differences.

**Figure 2 plants-11-03522-f002:**
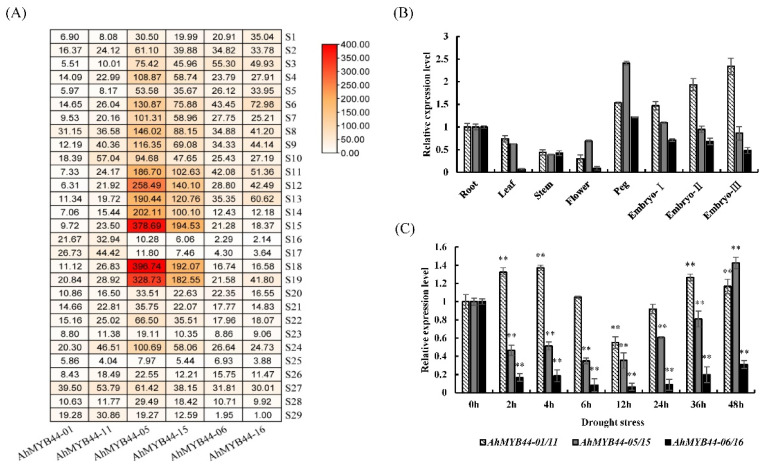
Expression patterns of *AhMYB44s* in tissues and responses to different stresses. (**A**). Transcription profiling of *AhMYB44s* in different tissues or organs of peanut and in leaves after different stress treatments. S1: cotyledon; S2: leaf; S3: florescence; S4: gynophore; S5: root; S6: root stem junction; S7: root tip; S8: root nodule; S9: stem; S10: stem tip; S11–S13: pericarps from pods after peg penetration into the soil for 10, 30, or 50 days, respectively; S14–S17: embryos from pods after peg penetration into the soil for 10, 20, 30, or 50 days, respectively; S18–S19: testas from pods after peg penetration into the soil for 40 or 60 days, respectively; S20: leaf treated with ddH2O; S21: leaf treated with abscisic acid (ABA); S22: leaf treated with salicylic acid (SA); S23: leaf treated with brassinolide (BR); S24: leaf treated with paclobutrazol (PBZ); S25: leaf treated with ethephon (EP); S26: leaf at room temperature; S27: leaf with low-temperature treatment; S28: leaf with normal irrigation; S29: leaf with drought treatment. (**B**). RT-qPCR analysis of relative expression levels of *AhMYB44s* in different peanut tissues. Embryos I–III: embryos from pods after peg penetration into the soil for 20, 30, or 50 days, respectively. (**C**). RT-qPCR analysis of relative expression levels of *AhMYB44s* in the leaves of peanut seedlings at four-leaf stage under drought treatment. Data are presented as means and standard deviations of three independent experiments. Aster-isks indicate statistical difference (** *p* < 0.01) compared to WT.

**Figure 3 plants-11-03522-f003:**
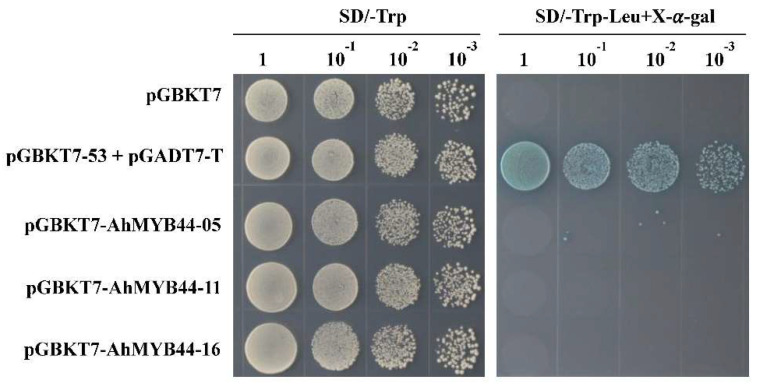
Transcriptional activation analysis of AhMYB44s in yeast. The recombinant vectors pGBKT7-AhMYB44-05/11/16 were transferred into yeast strain AH109. Their transcriptional activation ability was analyzed by growth on SD/-Trp and SD/-Trp-Leu with X-α-gal plates. pGBKT7: negative control; pGBKT7-53+pGADT7-T: positive control.

**Figure 4 plants-11-03522-f004:**
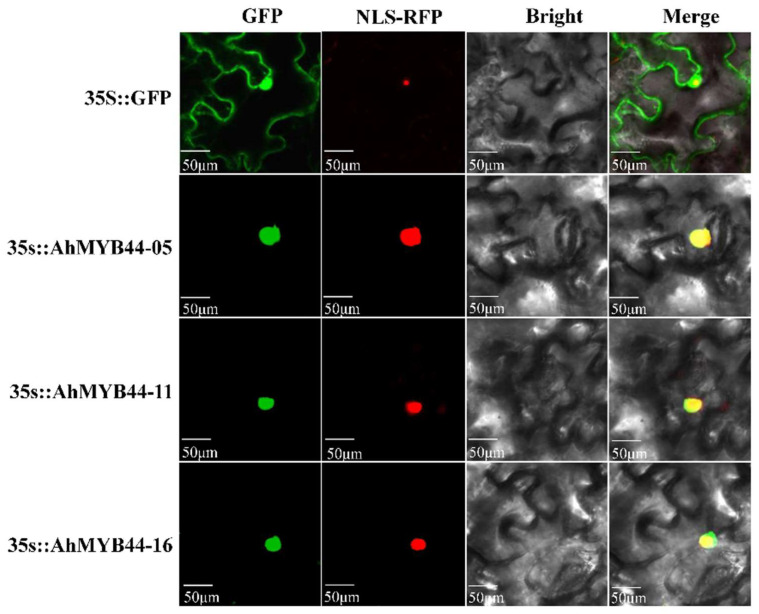
Subcellular localization of AhMYB44s in tobacco leaves. *Agrobacterium* carrying the *35S:AhMYB44-05/11/16: GFP* and *35S: GFP* genes were infiltrated into *N. benthamiana* leaves. Nucleus-localized NLS-RFP fusion protein was used as a nuclear localization marker. Images were obtained using confocal microscopy 2 d after agroinfiltration. Bars = 50 μm.

**Figure 5 plants-11-03522-f005:**
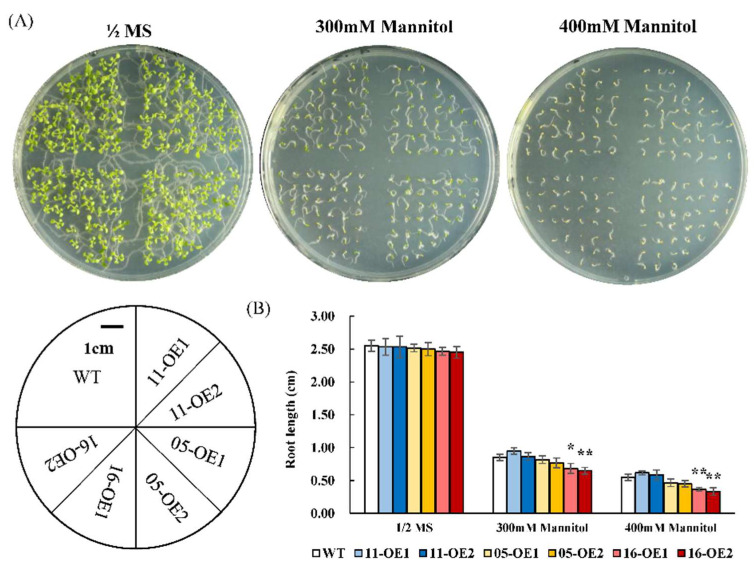
Analysis of drought tolerance at seed germination stage. (**A**) Morphology of WT and transgenic seeds germinated for 8 days on half-strength MS medium with 0, 300, or 400 mM mannitol. Below, the distribution diagram for the WT and transgenic lines on the culture plate is shown. Bars = 1 cm. (**B**) Root length of 8 day old WT, 11-OE1/OE2, 05-OE1/OE2, and 16-OE1/OE2 plants. Data are presented as means and standard deviations of three independent experiments. Asterisks indicate statistical difference (* *p* < 0.05, ** *p* < 0.01) compared to WT.

**Figure 6 plants-11-03522-f006:**
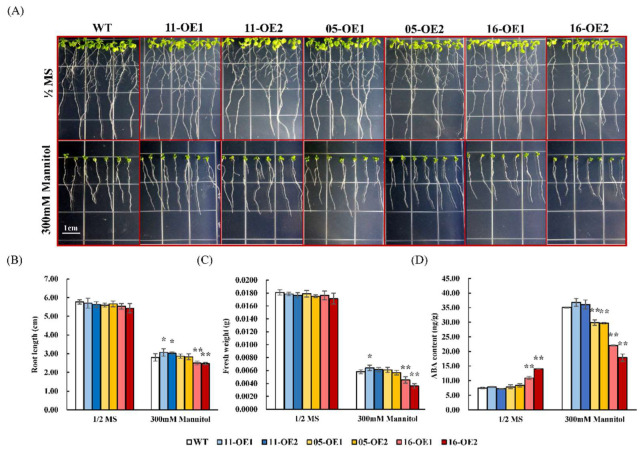
Analysis of drought tolerance at the seedling stage. (**A**) Morphology of WT and transgenic seedlings growing for 7 days on half-strength MS medium with 0 or 300 mM mannitol. Bars = 1 cm. (**B**–**D**) Root length, fresh weight, and ABA content of WT, 11-OE1/OE2, 05-OE1/OE2, and 16-OE1/OE2 plants. For fresh weight analysis, each repeat contained the weight of three seedlings. Data are presented as means and standard deviations of three independent experiments. Asterisks indicate statistical difference (** p* < 0.05, ** *p* < 0.01) compared to WT.

**Figure 7 plants-11-03522-f007:**
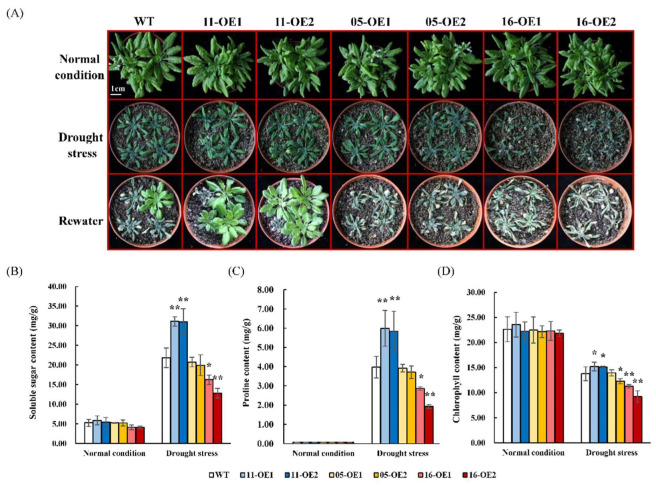
Analysis of drought tolerance in the adult stage. (**A**) WT and transgenic plants were grown in soil with sufficient water for one month before water was withheld for 15 days, after which the plants were allowed to recover for 7 days. Bars = 1 cm. (**B**–**D**) The proline, soluble sugar, and chlorophyll contents in the WT, 11-OE1/OE2, 05-OE1/OE2, and 16-OE1/OE2 plants. Error bars represent SDs for three independent replicates. Asterisks indicate statistical difference (** p* < 0.05, ** *p* < 0.01) compared to the WT.

**Figure 8 plants-11-03522-f008:**
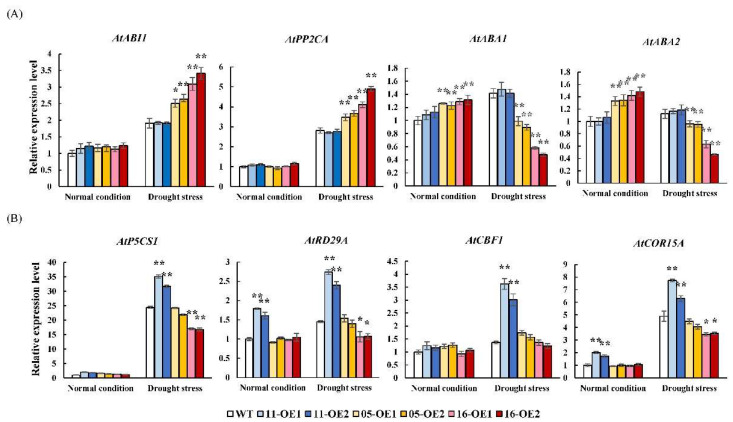
RT-qPCR analysis of stress-related genes in WT and transgenic *Arabidopsis*. Total RNA was isolated from the leaves of transgenic and WT plants after 3 days under stress. The *Atactin2* gene was used as a reference gene. The expression levels of genes in the WT plants under normal conditions were defined as “1”, and the 2^−ΔΔCT^ method was used to evaluate relative expression. (**A**) Relative expression levels of ABA-related genes: *AtABI1*, *AtPP2CA*, *AtABA1*, and *AtABA2*. (**B**) Relative expression levels of stress-response genes: *AtP5CS1*, *AtRD29A*, *AtCBF1*, and *AtCOR15A*. Data are presented as means and standard deviations of three independent experiments. Asterisks indicate statistical difference (* *p* < 0.05, ** *p* < 0.01) compared to the WT.

**Table 1 plants-11-03522-t001:** General information about *AhMYB44s*.

Name	mRNA ID in NCBI	Corresponding Sequence in PeanutBase	Chromosome No.	ORF		Amino Acids		MW (KDa)	PI
				bp	Homology (%)	aa	Homology (%)		
*AhMYB44-01*	XM_025841764	-	Arahy.01 (A1)	948	98.3	316	99.1	34.46	9.57
*AhMYB44-11*	XM_025771501	-	Arahy.11 (B1)	948		316		34.37	9.57
*AhMYB44-05*	XM_025842871	*Arahy.HL1WF3*	Arahy.05 (A5)	1056	98.8	352	99.7	37.29	9.79
*AhMYB44-15*	XM_025796710	*Arahy.CPTS0I*	Arahy.15 (B5)	1056		352		37.32	9.79
*AhMYB44-06*	XM_025751114	-	Arahy.06 (A6)	759	98.9	253	98.4	27.64	8.77
*AhMYB44-16*	XM_025804513	-	Arahy.16 (B6)	759		253		27.58	8.98

**Table 2 plants-11-03522-t002:** Motif elements potentially associated with the stress response of *AhMYB44s*.

Motif	(+: Forward Sequence, −: Reverse Sequence) Distance from ATG						Function
	*AhMYB44-01*	*AhMYB44-11*	*AhMYB44-05*	*AhMYB44-15*	*AhMYB44-06*	*AhMYB44-16*	
ARE (AAACCA)	(−) 173	(−) 174, (+) 975	(−) 144, (−) 919	(−) 144, (−) 920	(−) 135, (−) 974	(+) 573, (−) 995	Cis-acting regulatory element essential for anaerobic induction
CGTCA	(+) 325, (+) 735	(+) 326	(+) 312, (−) 719	(+) 312, (−) 720	(−) 924	(−) 945	Cis-acting regulatory element involved in MeJA responsiveness
MBS (CAACTG)	(+) 421	(+) 422, (+) 806	-	-	-	-	MYB binding site involved in drought inducibility
TATC-box (TATCCCA)	(−) 300	(−) 301	(−) 287	-	-	-	Cis-acting element involved in gibberellin responsiveness
GARE (TCTGTTG)	-	(+) 620	-	-	-	-	Gibberellin-responsive element
GC motif (CCCCCG)	-	(+) 717	(+) 1247	(+) 1248	-	-	Enhancer-like element involved in anoxic-specific inducibility
O2 site (GATGATGTGG/ GATGACATGG)	-	(−) 1130	(−) 1132	(−) 1133	-	-	Cis-acting regulatory element involved in zein metabolism regulation
ABRE (ACGTG/ CACGTA)	-	-	(−) 588, (−) 811, (−) 1182, (+) 1183, (−) 1346	(−) 592, (−) 812, (−) 1183, (+) 1184, (−) 1347	(−) 266	(−) 278	Cis-acting element involved in abscisic acid responsiveness
LTR (CCGAAA)	-	-	(−) 1292	(−) 1293	-	(−) 1265	Cis-acting element involved in low-temperature responsiveness
TC-rich repeats (GTTTTCTTAC)	-	-	(−) 840	(−) 841	-	-	Cis-acting element involved in defense and stress responsiveness

## Data Availability

Not applicable.

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
