# Peer review of "Identification of Peanut AhMYB44 Transcription Factors and Their Multiple Roles in Drought Stress Responses"

_plants, 2022, doi:10.3390/plants11243522_

Round 1

Reviewer 1 Report

In this manuscript, the authors completed a comprehensive analysis of peanut MYB44 memebers for their expression, properties and functions. Overall, this work presents a nice result regarding the roles of MYB44 genes in response to drought stress. However, minor revisions still require to be done before publication.

1.  In the introduction section, I cannot confirm whether all mentioned MYB44 genes from diverse plant species belong to the same subgroup 22?

2.For Figure 1, a phylogenetic tree with more MYB members should be provided to ensure the subgroups of AhMYB44 memebers.

3.For Figure 4, what indicated by RFP?

4. Scare bars can be provided for some figures.

5. Since these MYB44s may function as repressors, the biological functions of these MYB44s in the regulation of the stress-related genes (e.g. Figure 8) can be further discussed.

6. Overall English writing is good but still can be further improved in the text.

Reviewer 2 Report

Dear Authors 

I appreciate the idea and effort. The research designed very and performed very good. However, as you also have in your citation, Chen et al (2014), already identified 30 MYB gene family in peanut (https://doi.org/10.1016/j.gene.2013.08.092). You identified MYB families in peanut using Arabidopsis MYB sequence as query and it is perfect that you evaluated their role in Arabisopsis transgenic mutants. Now I am wondering why didn't you used the MYB genes that already identified by Chen et al. (2014)? Are the MYB genes that you identified same as what Chen identified before? You even didn't include them in phylogenetic tree. Please make it clear in your manuscript that:

Are the genes that you identified different than what Chen et al, (2014) identified?

Use all the genes from Chen in your phylogenetic analysis. 

Best wishes

Author Response

Dear reviewer,

Thank you for your suggestion. Since the purpose of our study is to understand the function of MYB44 homologous genes in peanut, we used AtMYB44 that more studied as a query. The problem you mentioned is really that we neglected, and we have supplemented this part. The result showed that AhMYB44-01 is a same protein with AhMYB4, and also AhMYB44-16 as AhMYB3. We modified the phylogenetic analysis by adding nine R2R3-MYBs from Chen. Why is 9 not 30. In Chen et al (2014), 30 MYB gene: one R1R2R3-MYB, 9 R2R3-MYB and 20 MYB-related members. For AhMYB44s are R2R3-MYB TFs. So just compare them with 9 R2R3-MYB.

Reviewer 3 Report

The authors have explored the functions of MYB transcription factor in peanut with six AhMYB44 genes identified in cultivated peanut with different functions under drought stress by performing several well-established methodologies. I believe that the authors have provided sufficient background, explained well the experimental strategies for reproducibility, presented the results with appropriate tables and figures, and concluded appropriately based on data available. I have no major technical concerns but some minor revisions, including some grammatical and editorial errors, i.e., appropriate spacing is needed for numerous cases of quantitative presentations, e.g., it should be “300 mM” but not “300mM” and it should be “2 h” but not “2h” throughout the entire manuscript. Therefore, I would recommend that the authors carefully proofread the entire manuscript for consistency and accuracy. Some of these revisions are listed here for the authors to consider if a revision is requested by the editor but again, there are more of these errors throughout the entire manuscript.

Introduction:

Line 55: “subgroup 22” is abbreviated as “S22” but then “subgroup S22” or “S22 subgroup” are not appropriate anymore, please be consistent with the use of this throughout the entire manuscript.

Lines 88-90: logically confusing sentence, please rewrite.

Results:

Line 107: in table 1, the URL or reference of peanut database should be provided.

Line 123: in table 2, the “(stand)” +, -, should be explained.

Line 126, change “phylogenetic tree analysis” to “phylogenetic analysis”

Figure 1: resolution should be improved…part A is hardly seen…no need to box 6 names…

Figure 4: line 204, “genes was”??

Figure 5: if part B is used to show the three plates in part A, then part B should be displayed prior to part A or part of part A.

Figures 6 and 7: no need to box the top and left labels.

Materials and methods:

Line 382, the authors may explain the selection of PEG6000 of 20%, any pre-experiments done? because it seems that PEG6000 is used at different concentrations in the literature.

Line 416: the authors never gave the full name of the RTq-PCR, is this the same as the “semi-quantitative reverse transcription PCR” (line 457)??

Discussion:

I would recommend that the authors establish 2-3 subsections so the discussion could be more focused.

Lines 290-291, please correct the grammar.

Line 333, please correct “As we known some……”

Round 2

Reviewer 2 Report

Dear Authors

Thank you so much for your reply and addressing the problem. There are minor correction in manuscript that would be better to be corrected. 

1-Line 148: You should mentioned about supplementary file in the main text and not in the figure legend. 

2- Line 175: Showed upregulated. It should be showed upregulation or simply upregulated. 

3- Line 220: It might be better to write 5, 11 and 16 instead of  5/11/16

Best wishes

Author Response

Dear reviewer,

Thank you for reviewing our manuscript carefully, we modified it according to your opinions.

1-Line 148: You should mentioned about supplementary file in the main text and not in the figure legend.

  Answer: Modified. Supplementary table S1 were added in Materials and methods 4.2.

2- Line 175: Showed upregulated. It should be showed upregulation or simply upregulated.

  Answer: Modified.

3- Line 220: It might be better to write 5, 11 and 16 instead of 5/11/16

  Answer: Modified.